# SALMONN-omni: A Speech Understanding and Generation LLM in a Codec-free Full-duplex Framework

## Abstract

Speech large language models (LLMs) offer a unified approach to handling various speech-processing tasks using a single autoregressive model built on discrete speech and audio codecs. Unlike traditional pipeline-based systems, which involve separate components for speech recognition, understanding, and generation, end-to-end speech LLMs can capture both verbal and non-verbal information, such as paralinguistic and speaker characteristics. This enables full-duplex capabilities, allowing the system to listen and speak simultaneously with low latency, making it ideal for conversational AI. In this paper, we introduce a novel codec-free, full-duplex framework for speech understanding and generation, and present SALMONN-omni, an instance of this speech LLM. SALMONN-omni can listen to its own generated speech and background sounds while speaking. To align the frame rate gap between text and audio, we propose a novel *thinking* step, ensuring high performance on pre-trained tasks. Using a two-stage *understand then generate* training approach, SALMONN-omni effectively addresses a variety of streaming speech tasks, including speech recognition, synthesis, enhancement, dereverberation, target speaker extraction, and spoken question answering.

## 1 Introduction

Large language models (LLMs) have established a new approach to problem-solving and task execution through natural conversations. Speech, being a fundamental form of human communication, acts as an intuitive and effective means for interactions between humans and LLMs. As a result, there is a growing research emphasis on enhancing the spoken input and output capabilities of LLMs. Some recent studies have focused on equipping LLMs with a comprehensive understanding of speech and audio, such as SALMONN (Tang et al., 2024; Sun et al., 2024; Yu et al., 2024) and LTU (Gong et al., 2024; 2023), while other research has explored utilizing LLMs' advanced language understanding abilities to develop more sophisticated speech generation and processing methods (Hao et al., 2023).

To further advance the naturalness of interaction with LLMs, more recently, full-duplex speech LLMs have been developed that support both speech understanding and generation. Some work achieves this by integrating inputs or outputs of standalone speech recognition and synthesis systems into LLMs (Wu et al., 2024; Huang et al., 2024). However, cascaded systems transcribe, understand and generate in series, resulting in systematic error accumulation and high latency, impeding the fluidity of real-time conversations with users. Meanwhile, end-to-end speech LLMs have been investigated, which often discretize speech into tokens and extend the LLM's vocabulary to support speech input and output (Ma et al., 2024; Zhang et al., 2023a; Défossez et al., 2024; Rubenstein et al., 2023). While these models achieve lower latency and end-to-end training, tokenized speech representations limit the expressivity of the high-dimensional speech signal due to the constraints on the number of tokens, often resulting in suboptimal performance in speech modelling.

This paper proposes SALMONN-omni, a codec-free full-duplex framework for low-latency streaming speech understanding and generation with speech LLMs. SALMONN-omni enables speech production and the perception of surrounding sounds and its own speech at the same time. In contrast to existing methods which rely on specific tokenization of the speech signal, SALMONN-omni models

speech features in a continuous space that is independent of any specific discrete tokenizations or audio codecs. Streaming speech encoders and generation modules are connected to SALMONN-omni via a streaming cross-attention structure. Moreover, a "turn-taking" mechanism is proposed in SALMONN-omni which enables the prediction of when to start a turn-taking conversation, enhancing the seamless speech-based human-AI interaction.

- This paper proposes SALMONN-omni, the first codec-free full-duplex speech-LLM that supports low-latency speech understanding and generation, enabling natural and spontaneous speech-based human-AI interactions.
- SALMONN-omni employs continuous-space speech representations without relying on discrete tokenizations or audio codecs. This avoids the loss of information during quantization as well as being compatible with any downstream speech generation systems.
- SALMONN-omni is the first to incorporate a streaming cross-attention module in full-duplex LLM, supporting highly efficient streaming inputs and outputs.
- SALMONN-omni further introduces a "turn-taking" mechanism to predict when to start a turn-taking conversation, improving the seamless interaction.

## 2 RELATED WORK

With the recent research advancements in multimodal LLMs, LLMs have been used for both speech and audio understanding and generation. SALMONN(Tang et al., 2024), Qwen-audio(Chu et al., 2023) and LTU(Gong et al., 2024; 2023) are early investigations that demonstrated generic audio understanding abilities with LLMs, which significantly broadened the scope of tasks a single model can perform. Later work further exploits the power of specific tasks such as speech translation (Chen et al., 2023b), entity retrieval (Wang et al., 2023) or emotion recognition (Latif et al., 2023), or to improve specific aspects such as task overfitting (Deng et al., 2024) or data efficiency (Katsumaru et al., 2009; Manakul et al., 2024), etc. On speech and audio generation, LLMs have either been used to provide better textual descriptions that facilitate text-to-speech (TTS) synthesis (Zhang et al., 2023b; Leng et al., 2023), or been used to provide tokens that can directly be mapped to audio (Dekel et al., 2023; Wu et al., 2023). In particular, Dekel et al. (2023) studies streaming speech generation alongside text generation, enabling seamless spoken response generation.

Full-duplex speech LLMs have recently become a research focus, with various methods being proposed that enable both speech understanding and generation simultaneously. AudioGPT (Huang et al., 2024) and NextGPT (Wu et al., 2024) are examples where separate speech recognition and synthesis systems were integrated to enable speech-based interaction with LLMs. More recently, researchers have investigated end-to-end trainable speech-text interfaces with LLMs by expanding the LLM vocabulary with speech tokens representing different speech signals (Ma et al., 2024; Zhang et al., 2023a; Défossez et al., 2024; Rubenstein et al., 2023), which suffers from the trade-off between number of tokens and representation ability. In contrast to these methods, SALMONN-omni is the first speech LLM that directly leverages the continuous speech representation space that is independent of any specific sets of speech tokens.

## 3 CODEC-FREE FULL-DUPLEX SPEECH UNDERSTANDING AND GENERATION FRAMEWORK

A full-duplex speech understanding and generation framework must address four key challenges. First, it should support streaming speech input and output. Second, it must provide a mechanism to handle both input and output streams simultaneously. Third, it must incorporate a period to synchronize the states of the input and output streams. Finally, it should implement a strategy for the model to learn turn-taking in natural human conversations, such as when to backchanneling or to be badged in by the user.

Instead of tokenizing speech into discrete codecs and using *next-token-prediction* to modelling both the textual and auditory tokens, we propose the first codec-free full-duplex speech understanding and generation framework, as shown in Figure 1, which keeps the LLM generating only text tokens to avoid jointly modeling tokens of two modalities in a single sequence model. Four key features

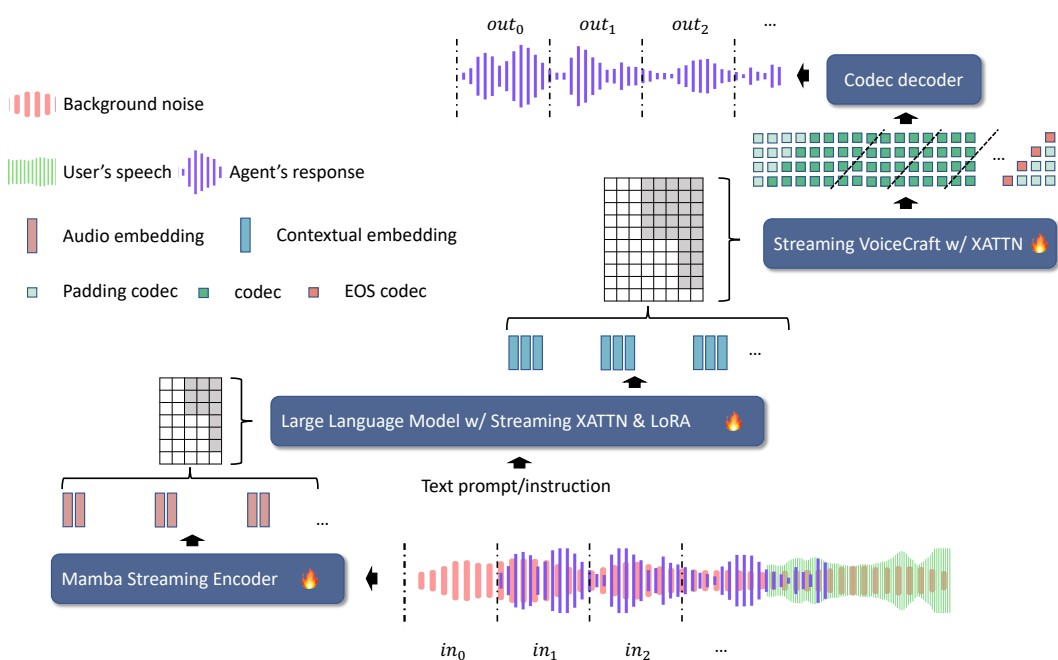

Figure 1: The structure of SALMONN-omni implemented in the proposed codec-free full-duplex speech understanding and generation framework.

in our framework address the challenges for implementing a full-duplex system while keeping the core of the model codec-free. First, we propose to incorporate the LLM with a streaming speech encoder and a streaming speech synthesizer to enable streaming speech input and output. Second, the speech encoder and synthesizer are connected to the LLM with several cross attention layers (Vaswani et al., 2017) so that the model can handle both input and output streams simultaneously in an autoregressive manner. Third, we set a fixed period for the synchronization between input and output streams. Specifically, a fixed policy is introduced for the mapping between the number of text embeddings and the duration of the speech streams. Finally, a novel "*thinking*" strategy is proposed to enable the model to learn turn-taking in natural human conversations. As shown in Figure 2, when the model should not response to the inputs, we just send "thinking" tokens into the LLM and train the model not to generate *start-of-generation* token. Because we don't force the model to generate any specific token, the LLM will not tend to bias to one special token and the degradation of the performance can be negligible. Moreover, due to the frame rate difference between text and speech, even if the textual embeddings are finished generating, we still send "thinking" embeddings into the speech synthesizer to make sure the full-duplex framework is complete.

## 4 METHODOLOGY

### 4.1 MAMBA STREAMING ENCODER

We introduce the Mamba (Gu & Dao, 2023) streaming encoder to extract continuous embeddings from speech inputs. Following Yang et al. (2024), multi-teacher knowledge distillation is employed to align the feature space of the streaming encoder with those of teacher models, using only unlabeled data. Fbank features are extracted at a frame rate of 100Hz, after which two convolutional layers downsample the features to 50Hz. Two adjacent embeddings are then concatenated into one embedding which are used as the input to a series of standard Mamba language model blocks. The generated speech features $\mathbf{S}$ are used to calculated the multi-teacher knowledge distillation loss as

$$loss_{MTL} = \lambda_1 * loss_{ASR} + \lambda_2 * loss_{AT} \tag{1}$$

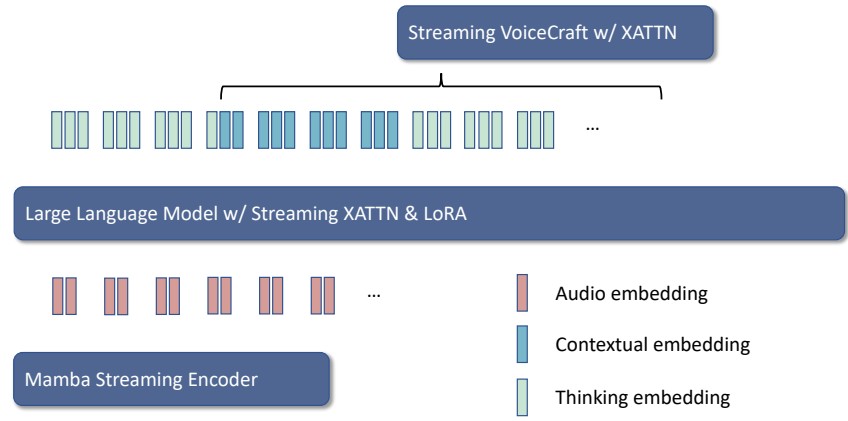

Figure 2: The proposed "thinking" strategy, which enable SALMONN-omni to learn the turn-taking phenomena in human natural conversation.

Here we consider two teachers Whisper-large-v3 (Radford et al., 2023) as the teacher for speech recognition (ASR) and BEATs (Chen et al., 2023a) as the teacher for audio tagging (AT).

### 4.2 STREAMING TTS

Our streaming TTS system builds on the popular open-source TTS VoiceCraft-830M (Peng et al., 2024), which employs a codec language model architecture. A streaming TTS is capable of streaming generation conditioned on a streaming increased input. A codec language model is suitable for the requirement of streaming generation, since it generates codec codes autoregressively, which can be transformed to speech waveform via a codec decoder. However, current codec language models require the entire text sequence to generate codec codes, limiting their ability to accept incremental text input during generation. To address this limitation, we have implemented several modifications that allow our model to accommodate streaming input effectively.

Our solution involves transforming a non-causal mask decoder-only codec language model into a causal cross-attention decoder model, as shown in Figure 3. Traditional decoder-only codec language models struggle to process incremental text input because text and codec code are combined as the decoder input. Once the model begins generating codec codes based on the entire text sequence, it can no longer accept new text input, or the codec code generation would be interrupted. To overcome this limitation, we implement a cross-attention decoder architecture that separates text input from codec code input. The embeddings of generated tokens from large language models serve as the text input, fed to the cross attention, while the codec codes function as the decoder input. A stack of linear layers is employed to transform the embeddings to the dimension of Voice-Craft attention. To effectively model streaming incremental input, we utilize a fixed-number causal cross-attention; for instance, generating 20 codec codes from 2 embeddings provided by the large language model. This causal cross-attention mechanism enables the codec language model to generate audio seamlessly based on streaming input.

### 4.3 TRAINING STRATEGY

The *understand then generate* training strategy is utilized to enable SALMONN-omni with streaming speech understanding and generation abilities, as illustrated in Figure 4. The training loss may contain two parts: the text LLM loss $loss_{LLM}$ which is the cross entropy loss between the text tokens and corresponding labels and the speech TTS loss $loss_{TTS}$, the cross entropy loss between the codec codes and corresponding labels. The weight of different losses may be changed across different training stages.

$$loss = w_{text} * loss_{LLM} + w_{speech} * loss_{TTS} \qquad (2)$$

The first stage is understanding training, aligning speech encoder to the LLM to equip it with speech understanding ability. Speech encoder, cross attention module and LoRA in LLM are trained on

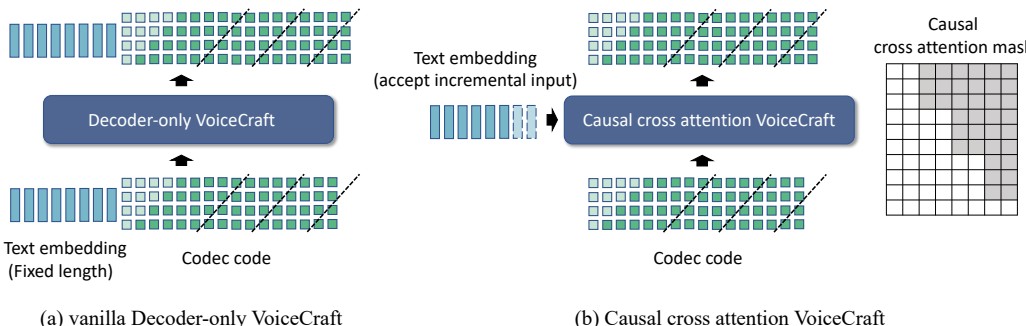

(a) vanilla Decoder-only VoiceCraft      (b) Causal cross attention VoiceCraft

Figure 3: The sketch map of our streaming TTS module, highlighting two key modifications. Cross attention enables the model to accept incremental input when generating codec codes. Causal cross attention simulates the streaming increased text input during training.

all the speech understanding tasks, including streaming ASR, noisy ASR, target speaker ASR and speech QA.

The second stage is generation training, enabling the speaking ability of the model. Two generation training strategies are explored in this paper. The first strategy involves training the LoRA in both the LLM and VoiceCraft across all generation tasks, using a combination of text LLM loss and speech TTS loss. The weight of text LLM loss $w_{text}$ is 0.1 while The weight of speech TTS loss $w_{speech}$ is 1. Notably, we modify the zero-shot TTS task into a zero-shot continual TTS task during training, retaining only one percent of the original zero-shot TTS data. This adjustment is necessary because the generated text also appears in the text prompt, which could lead the TTS model to bypass learning shortcuts by attending directly to the text prompt, instead relying on the proper relationship between the generated text and the corresponding speech. The another option of generation training is to freeze the LLM and only train VoiceCraft only on zero-shot continual TTS task. Here $w_{text}$ is 0 and $w_{speech}$ is 1. The speech generation ability learned by zero-shot continual TTS task can be generalized to other generation tasks, like SE, dereverberation and TSE.

We demonstrate the proposed "thinking" strategy with understanding tasks and train the model to predict the time for turn-taking. For simplicity, we set the ending point for each speech utterance as the turn-taking point.

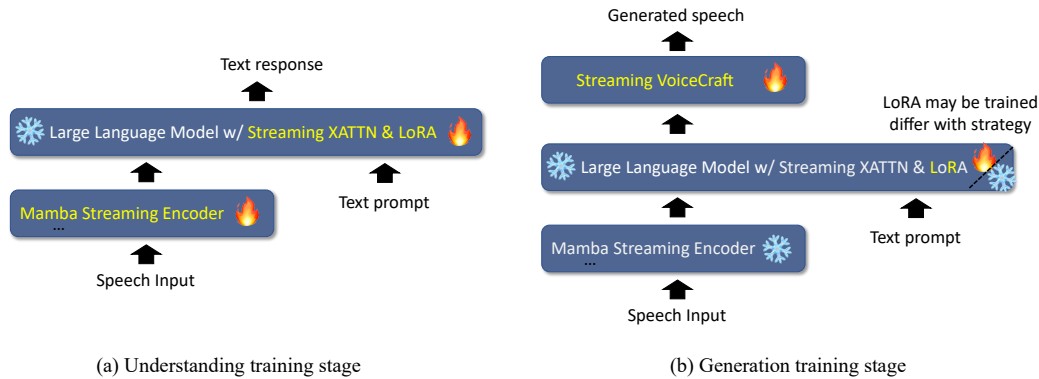

(a) Understanding training stage      (b) Generation training stage

Figure 4: The *understand then generate* training strategy of our method. First training the speech encoder, cross attention module and LoRA in LLM to align speech modality to LLM. Then training the streaming TTS to enable streaming speech generation based on text token embeddings of LLM.

## 5 EXPERIMENTAL SETUP

### 5.1 TASK CONFIGURATION

SALMONN-omni is a multi-task speech generation model that supports zero-shot TTS, speech enhancement, dereverberation, and target speaker extraction. All generation tasks are formulated as a "Understand then Speak" manner, as demonstrated in Fig 1. The speech LLM generates text based on the text prompt and audio prompt, then the streaming TTS produces speech from the embeddings of the generated tokens synchronously. The generated speech is fed back to the speech encoder during generation process. Additionally, a 3-second speaker prompt is placed at the beginning of Voicecraft to control the voice of the generated speech. Since our primary focus is on extracting meaning and producing clear speech—rather than reconstructing speech with all its subtle details—we define our tasks more precisely as zero-shot TTS, respeaking SE, respeaking dereverberation, and respeaking TSE.

Table 1: Configuration of different speech generation tasks.

| Task | Audio prompt | Generated text | Generated speech |
|---|---|---|---|
| zero-shot TTS | speaker prompt | the text to be generated | the speech to be generated |
| respeaking SE | degraded speech | noisy ASR result | respeaking clean speech |
| respeaking dereverb. | degraded speech | noisy ASR result | respeaking clean speech |
| respeaking TSE | mixed speech, speaker prompt | target speaker ASR result | respeaking target speaker speech |

### 5.2 DATA

We utilize Libriheavy Kang et al. (2024) as the basic dataset for our experiment. Only speech segments under 10s, which are approximately 2.7M, are used for generation tasks, since the training is unstable when there is a lot of long speech segments. For continual TTS, the previous text is generated by Llama-3-8B-Instruct Dubey et al. (2024) given the text to generate. For respeaking SE, the speech segments are degraded following the setting of DNS Challenge 2023 Dubey et al. (2023) with a random SNR from 5 to 20. For respeaking dereverberation, the speech segments are degraded following the setting of DNS Challenge 2023 Dubey et al. (2023) with the impulse response from Ko et al. (2017). For respeaking TSE, two speech segments are mixed with a random overlapping ratio from 0.1 to 0.5. For continual TTS, all speech segments under 10s are used, While for SE, dereverberation and TSE, 1M speech segments are randomly selected to conduct the training datasets. The train, valid and test splits are generated from the large, dev, test_clean spilts of Libriheavy, respectively. For automatic speech recognition (ASR) task, we also include widely used LibriSpeech 960h dataset as training data besides the extracted 2.7M segments from LibriHeavy. VoiceAssistant-400k (Xie & Wu, 2024) dataset is used for spoken question answering task but we only keep the samples in `qa_assistant_v1_7k` and `alpaca_gpt4_en_55k` categories and regenerate the answers for these questions with GPT-4o-mini. For the pretraining of the Mamba streaming encoder, GigaSpeech-XL (Chen et al., 2021) subset and AudioSet (Gemmeke et al., 2017) are used.

### 5.3 EVALUATION

For understanding tasks, word error rate (WER) is used for ASR task. For the Spoken QA task, we use GPT-4o-mini to judge whether the answer generated is suitable for answer the question and we report the success rate.

For generation tasks, objective predicted mean opinion score (MOS) and speaker similarity (SIM) are reported. We utilize UTMOS Saeki et al. (2022)[1] as our MOS prediction module, which can estimate an objective score of MOS to evaluate the speech naturalness. For SIM, the pre-trained speaker verification model WavLM-TDCNN Chen et al. (2022)[2] is used to estimate the similarity between the generated speech and the reference speech.

---

[1] https://github.com/tarepan/SpeechMOS
[2] https://huggingface.co/microsoft/wavlm-base-plus-sv

# 6 RESULTS

## 6.1 UNDERSTANDING

| Strategy | test clean | test other | Spoken QA |
|:---:|:---:|:---:|:---:|
| 1 | 6.3% | 11.1% | 38.5% |
| 2 | 6.0% | 10.5% | 40.5% |

Table 2: Results of the speech recognition and spoken QA tasks.

As shown in Table 2, both strategies 1 and 2 can perform speech recognition and spoken question answering tasks. Strategy 2 performs slightly better than strategy 1 because only VoiceCraft is finetuned and keeps the performance after the first stage.

## 6.2 GENERATION

| Strategy | embedding layer | TTS | | SE | | dereverberation | | TSE | |
|:---:|:---:|:---:|:---:|:---:|:---:|:---:|:---:|:---:|:---:|
| | | MOS | SIM | MOS | SIM | MOS | SIM | MOS | SIM |
| | 0 | 3.31 | 0.86 | 3.36 | 0.88 | 3.31 | 0.86 | 3.38 | 0.86 |
| 1 | 16 | 3.47 | 0.87 | 3.61 | 0.92 | 3.61 | 0.92 | 3.59 | 0.91 |
| | -1 | 3.51 | 0.87 | 3.61 | 0.91 | 3.59 | 0.91 | 3.52 | 0.90 |
| | 0 | - | - | 3.47 | 0.86 | 3.45 | 0.86 | 3.52 | 0.86 |
| 2 | 16 | - | - | 3.53 | 0.88 | 3.54 | 0.88 | 3.59 | 0.89 |
| | -1 | - | - | 3.23 | 0.86 | 3.20 | 0.85 | 3.29 | 0.85 |

Table 3: Results of generation tasks on Libriheavy test-clean subset.

The performance of SALMONN-omni across various generation tasks is presented in Table 3. Results for the TTS tasks are omitted, as the LLM struggles to repeat the text for generation. The results demonstrate that SALMONN-omni is capable of generating clear, natural speech with a similar speaker voice defined by the speaker prompt, across different tasks. We also examined the impact of using embeddings from different LLM layers. In strategy 1, where both LoRA and Voice-Craft are fine-tuned, the later layers of the LLM perform better, as more parameters in LLMs can be finetuned to generate embeddings that VoiceCraft can interpret more effectively. In contrast, for strategy 2, where the LLM is frozen and only VoiceCraft is fine-tuned, the earlier layers prove more useful, as they retain more information from the input text.

## 6.3 TURN-TAKING

| Model | test clean | test other | Spoken QA |
|:---:|:---:|:---:|:---:|
| SALMONN-omni | 5.3% (100%) | 11.2% (99.6%) | 38.5% (94.6%) |

Table 4: Results for the speech recognition and spoken QA tasks demonstrate that the model effectively predicts turn-taking. The second set of numbers represents the success rate of the model in accurately determining the timing of turn-taking.

Table 4 shows that with the proposed "thinking" strategy, the model has a high success rate in predicting when to start generating. Moreover, because we set the end of the utterances as the turn-taking point, the speech recognition task turns into non-streaming recognition and the model performs better than the streaming one.

# 7 CONCLUSION

In this paper, we presented SALMONN-omni, a speech LLM built within a codec-free, full-duplex framework for speech understanding and generation. SALMONN-omni is capable of handling var-

ious streaming speech tasks, including automatic speech recognition (ASR), text-to-speech (TTS), speech enhancement (SE), dereverberation, and target speaker extraction (TSE). Additionally, we introduced a streaming Mamba encoder to facilitate real-time speech understanding and a causal cross-attention codec language model for effective streaming speech generation. Future work will focus on enhancing the stability and versatility of the framework, as well as exploring its potential for developing low-latency spoken dialogue systems.

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
