# OpenReview forum: "SALMONN-omni: A Speech Understanding and Generation LLM in a Codec-free Full-duplex Framework"
_ICLR.cc/2025/Conference — ICLR 2025 Conference Withdrawn Submission_

### Official Review · Reviewer_d4sH · 2024-10-31

**Soundness:** 2
**Presentation:** 2
**Contribution:** 2
**Rating:** 3
**Confidence:** 4

**Summary:**

This paper proposed a full-duplex LLM framework which can perform speech understanding and generation within the same system. The overall system includes the Mamba encoder to encode the input speech stream, the text-based LLM, and VoiceCraft-based Streaming TTS module. To enable the streaming feature, streaming Mamba is used for encoder module, and cross-attention is used for VoiceCraft module. Two stages of training are included, namely understanding and then generation with different training loss setting. For evaluation, a subset of LibriHeavy data set is used. WER results are reported for understanding task, while MOS scores and SIM scores are reported for generation tasks.

**Strengths:**

1, This work proposed to solve an important problem of end-to-end speech understanding and generation through the speech language model and TTS model.
2, Different methods are attempted.

**Weaknesses:**

1, Lack of innovation. The proposed end-to-end system is a combination of different existing methods.
2, One of the core parts of this work is the LLM. However, there is no description of the LLM module of this work at all. From the current version of the paper, the readers cannot tell if the LLM is pre-trained by the authors or if it is a public checkpoint from open-source work. Describing the LLM module is essential.
3, Cross-attention in both LLM and TTS modules seems to be one of the key components of this work. However, this part is not clearly described. Using equation is suggested for precise description.
4, Insufficient experiments. i) The experiment part only reports the result of the proposed method without comparing it to other baseline method. Such baseline methods could be either traditional speech enhancement/dereverberation methods, or LLM-based methods. ii) There is no ablation study in terms of the performance of each submodule. One example is to compare the performance of Mamba to other autoregressive architecture for the speech encoding. Or dropping the “thinking” token to see how the performance is degraded.
5, The proposed work focuses on a streaming framework. However, there is no system latency result reported which is necessary, e.g., how long of the look ahead of speech signal is required to process. It would be even better if an ablation study of the performance against different latency settings.
6, Concepts are confusing. The paper mentioned that a codec-free method is proposed. However, in the TTS module of VoiceCraft, codec codes are generated. It seems the codec-free only occurs in the encoding and speech understanding stage, but not in the TTS stage. To have a consistent presentation of each part of the paper, you need to either drop the “codec-free” or make it more specific to codec-free in understanding part.
7, Some of the evaluation metrics are not clearly described. One example is how you measure the turn taking accuracy. Can you describe it using an equation? The other example is how you measure the success rate of the spoken QA task. Can you also describe it either using an equation or cite a reference paper?
8, Writing needs to be improved. i) There are many typos in the paper.  One example, line 355-356, “both LoRA and VoiceCraft are fine-tuned” should be “both LLM and VoiceCraft are fine-tuned”. ii) More detailed explanation is needed for these Tables (Table 2 and 4) in terms of the evaluation metrics. For example, if the results are corresponding to speech recognition, it’s necessary to include WER in the table or list as a notation.

**Questions:**

1, Do you have a demo link?
2, In terms of the speech encoder module, have you tried other autoregressive model architecture besides Mamba? If so, how is the performance compared to Mamba?
3, Why do you select VoiceCraft for speech generation?
4, What is the overall model size of the end-to-end system?

---

### Official Review · Reviewer_rmnA · 2024-11-03

**Soundness:** 2
**Presentation:** 2
**Contribution:** 3
**Rating:** 5
**Confidence:** 2

**Summary:**

This paper presents SALMONN-OMNI, a codec-free, full-duplex speech large language model (LLM) designed for real-time speech understanding and generation. Specifically, it incorporates a Mamba speech encoder, a text LLM, and VoiceCraft, with architectural modifications to support streaming. The model is evaluated on ASR, Spoken QA, zero-shot TTS, speech enhancement (SE) , dereverbration and target speaker extraction (TSE).  Ablations show good strategies for fine-tuning the model for understanding tasks and its performance on speech generation tasks.

**Strengths:**

**Originality**: The architectural modifications enabling streaming in a full-duplex setup without relying on codecs are innovative.

**Quality**: The results demonstrate the effectiveness of the proposed method on downstream tasks, though the absence of baseline comparisons makes it difficult to fully assess its relative performance.

**Clarity**: While the core methodology is well-presented, some details appear to be missing, which may impact overall clarity.

**Significance**: The paper introduces a novel approach to addressing an emerging and important problem.

**Weaknesses:**

The paper highlights that the proposed model represents speech features in a continuous space without relying on codecs or discrete tokens; however, it remains unclear whether this design choice is optimal, as no comparative analysis has been provided.
An evaluation using cascaded systems (eg: Qwen2 with VoiceCraft) could provide valuable insights into performance.
There are few missing details on task formulation, model specifications (text LLM) and training details that can help improve paper clarity.

**Questions:**

1. Could the authors include comparisons with a baseline system, such as cascaded or tokenized speech LLMs, to better validate the performance gains in latency and accuracy?

2. How was the text LLM initialized—was it pre-trained, and if so, on what data?

3. What were the training hyperparameters, and what rank was used in the LoRA configuration?

4. What were the computational requirements for training the model, including the number of GPUs used?

5. Which task formulations require interruption, and how was it determined that the interruptions occurred at appropriate points?

---

### Official Review · Reviewer_Jb8D · 2024-11-03

**Soundness:** 2
**Presentation:** 1
**Contribution:** 2
**Rating:** 1
**Confidence:** 4

**Summary:**

This paper describes a unified speech language model, SALMONN-OMNI, that enhances speech generation capabilities (zero-shot TTS, speech enhancement, dereverberation, and target speaker extraction) by extending the SALMONN system. Compared with other speech-language modes, this method does not rely on discrete speech token representation and also natively supports streaming processing for both input and output. The experiments show their effectiveness in four speech generation tasks and their turn-taking capability.

**Strengths:**

- Unified speech language models have become a very hot topic following the emergence of GPT-4 O.
- Conventional methods have a lot of difficulties in using unified discrete speech representation (e.g., neural codec vs. speech SSL representations). This paper avoids using this representation (however, I could not understand their method).
- Streaming capabilities for both input and output are important for real-time speech interface.
- Showing the effectiveness of the proposed approaches in five tasks (zero-shot TTS, speech enhancement, dereverberation, target speaker extraction, and turn-taking).

**Weaknesses:**

- The paper has substantial issues in its presentation. Many explanations are either insufficiently detailed or lack internal consistency, and the descriptions of the figures are also lacking in detail. As a result, I struggled to understand much of the technical content of the proposed approach.
- The lack of reproducibility is a serious concern. Part of this stems from the presentation issues mentioned above, as the paper does not provide enough detailed explanations of the techniques to allow for the full reproduction of their methods. Furthermore, Section 5.2 does not include adequate information on how the training data is generated, particularly regarding SE, dereverberation, and TSE.
- The experimental results do not show a comparison with the other models, and it is difficult to justify their numbers.
  - At least Librispeech results in Table 2 are a way behind compared with streaming-based approaches (6.0 vs. 2.8 [1] in test_clean and 10.5 vs. 7.3 [1] in test_other).).


[1] Moritz, Niko, Takaaki Hori, and Jonathan Le. "Streaming automatic speech recognition with the transformer model." ICASSP 2020-2020 IEEE International Conference on Acoustics, Speech and Signal Processing (ICASSP). IEEE,

**Questions:**

- Section 3: In the phrase, "Third, it must incorporate a period to synchronize the states of the input and output streams," could you clarify what is meant by "states"? A more detailed explanation of these states would be helpful.
- Figure 1 and Related Explanations: The explanations surrounding Figure 1 are challenging to follow, especially the meaning of "LLM generating only text tokens." This is partly due to a lack of context on "contextual embedding." Please provide further details on this concept to improve clarity.
- Section 3, Token Generation Bias: The sentence, "Because we don’t force the model to generate any specific token, the LLM will not tend to bias to one special token and the degradation of the performance can be negligible," is difficult to interpret. Could you clarify this with examples? Specifically, what types of special tokens might cause biases, and how could these biases potentially degrade performance?
- Section 4.2, Explanation of VoiceCraft: The description in Section 4.2 is somewhat unclear. Additional explanations regarding the workings of VoiceCraft and how this method utilizes it would be beneficial.
- Table 1, Zero-shot TTS: Could you elaborate on why text generation is necessary in a zero-shot TTS context?
- Section 5.2, Use of Previous Text: The phrase, "For continual TTS, the previous text is generated," needs further clarification. Why is the generation of previous text necessary? A detailed explanation of this part would be helpful.
- Section 6.2, Intelligibility Test: Is there an intelligibility test included in the evaluation? Evaluating generated sound solely based on quality and speaker similarity may not be sufficient. Intelligibility is also an important factor.
- Table 3, Terminology: To avoid confusion, please refrain from referring to "MOS" in relation to DNN-based MOS in Table 3.
- Table 3, UTMOS Justification: Could you provide a rationale for using UTMOS, developed for TTS, in the context of SE, dereverberation, and TSE?
- Table 3, SE/Dereverberation/TSE Results: The results for SE, dereverberation, and TSE should include the performance of the original speech for comparison.
- Table 4, Success Rate Explanation: The explanation of the success rate is currently unclear. Could you provide additional details to clarify this measure?

---

### Official Review · Reviewer_wqsm · 2024-11-04

**Soundness:** 3
**Presentation:** 2
**Contribution:** 2
**Rating:** 5
**Confidence:** 3

**Summary:**

This work presents an extension of the "SALMONN" speech LLM, "SALMONN omni", specifically aimed at achieving a full-duplex streaming that can both take input from the user and generate output for the user simultaneously. The key claimed novelties described in the work consist of (1) the use of continuous embeddings instead of discrete tokens to represent the different modalities and (2) the use of cross-attention rather than self-attention. The draft describes a large amount of engineering effort as part of the overall system design. Good results are reported for the model, including ASR WERs and results for TTS MOS, speech enhancement, dereverberation, and target speaker extraction.

**Strengths:**

The use of speech LLMs performing a variety of speech-related tasks (ASR, TTS, speech enhancement, etc.), is a very hot topic, and models such as Moshi (referenced by the submission) exploring the concept of full duplex input/output for such models are of particular interest. This submission further explores this space, proposing some novel aspects, such as the use of embeddings rather than discrete tokens, and the use of cross-attention rather than self-attention, that are of high interest to the community. Notably, the work also makes use of the novel Mamba encoder, cast here in a streaming variant. Clearly, a lot of work went into this study. The references to prior work are overall good (though see below).

**Weaknesses:**

The primary weakness is a lack of clarity in the presentation. The model, architecture and loss functions are described conceptually more than mathematically, which in itself is not a bad thing, it's a bit hard to follow the wealth of methods and details described. Some abbreviations are not explained until the Conclusion; several aspects of the experimental evaluation are hard to follow.

Regarding the novelty of the work, I do think that, in the context of Speech LLMs, the use of cross-attention instead of self-attention is novel compared to the published studies I am aware of -- though the question of cross-attention vs self-attention in the context of speech LLMs is commonly discussed in the community. The submission presents arguments for the advantages of cross-attention, and demonstrates the practical viability of their architectural choices, but it does not compare head-to-head with the self-attentional version of their system to demonstrate experimentally some of those advantages. One could make the same comments about the use of continuous embeddings instead of discrete tokens. This, too, is novel (and interesting) compared to many published studies -- but the work AFIACT doesn't offer head-to-head comparisons of use of continuous embeddings vs discrete tokens.

The "understand then generate" strategy described in the submission is interesting, and is a key point of the study. However, the use of "thinking" tokens seems very similar to the use of empty/blank tokens in Seide et al. (2024), "Speech ReaLLM -- Real-time Streaming Speech Recognition with Multimodal LLMs by Teaching the Flow of Time", https://arxiv.org/abs/2406.09569, which I think should be cited.

Given that this is not the only full duplex Speech LLM system, the reader is left wondering which of the many architectural decisions are key differentiators, beyond the top-level ones (continuous embeddings vs discrete tokens, and cross-attention vs self-attention). Perhaps the work could be improved by offering a clearer overall contrast listing pros and cons w.r.t. e.g. the well-known Moshi model?

**Questions:**

Here are a number of comments/questions/suggestions about the text.

L103, "when to backchanneling or to be badged in by the user": fix grammar?

L107, "which keeps the LLM generating only text tokens to avoid jointly modeling tokens of two modalities in a single sequence model": improve grammar & phrasing?

L142, "model should not response to ..." --> "model should not respond to ..."

L144, "Because we don’t force the model to generate any specific token, the LLM will not tend to bias to one special token and the degradation of the performance can be negligible.": I could not quite understand what is meant here.

L145, " the textual embeddings are finished generating": fix grammar?

L155, re: teacher distillation, perhaps clarify the motivation here? When saying, "align the feature space", does "feature space" mean "extracted audio encodings", of Fbank features? I assume the former.

L157, "Two adjacent embeddings are then concatenated into one embedding": "embedding" here is a bit vague, I assume these are temporal frames of acoustic encodings/embeddings, and that was is being described is a 2x reduction in the frame rate -- or something like that?

L161, Eq. (1): I recommend using something like $\texttt{loss}_\texttt{MTL}$ etc., when a full word is part of an equation.

L187, "streaming increased input": improve grammar?

L189, "current codec LMs require the entire text sequence to generate code codes": why, exactly? Is the solution in Fig. 3 not applicable to a model using self-attention?

L201, "we utilize a fixed-number causal cross-attention; for instance, generating 20 codec codes from 2 embeddings": what does "fixed-number" refer to here...?

Figure 3: What do the diagonal dashed black lines in the figures represent?

L240, "... retaining only one percent of the original zero-shot TTS data.  This adjustment is necessary because the generated text also appears in the text prompt, which could lead the TTS model to bypass learning shortcuts by attending directly to the text prompt, instead relying on the proper relationship between the generated text and the corresponding speech." I cannot quite understand this key point, could you clarify? Also, perhaps this would be a good point to contrast the similarities & differences between the presented approach and other models in the literature, e.g. Moshi. I.e., How is the overall training scheme described in Section 4.3 similar or different from that used to train Moshi?

L245: SE and TSE: define these abbreviations here (I think they are only defined in the Conclusion)

L297: "when there is a lot of long speech segments": fix grammar.

L302: "..., While for ...": fix capitalization

L304: fix typo "spilts"

L314: "For the Spoken QA task, we use GPT-4o-mini to judge whether the answer generated is suitable for answer the question and we report the success rate." : could you add more details for how the success rate is defined?

Table 2: Could you specify exactly what Strategy 1 and Strategy 2 here refer to?

Tables 2 & 4: specify that the eval sets here are from LibriHeavy?

---

### Note · Authors · 2024-11-24

**Comment:**

We sincerely thank all the reviewers for their careful reviews and thoughtful feedback. We will incorporate your valuable suggestions to further improve our work.

**Withdrawal Confirmation:**

I have read and agree with the venue's withdrawal policy on behalf of myself and my co-authors.